# Updated determinations of $|V_{us}|$ with tau decays using new recent estimates of radiative corrections for light-meson leptonic decay rates

A. Lusiani[1*],

**1** Scuola Normale Superiore and INFN sezione di Pisa, Italy
* alberto.lusiani@pi.infn.it

January 13, 2022

16th International Workshop on Tau Lepton Physics (TAU2021),
September 27 – October 1, 2021

## Abstract

We update the $|V_{us}|$ determinations using the HFLAV 2018 report tau branching fraction results with recent new estimates of the $\pi\ell2$ and $K\ell2$ radiative corrections. There are minor changes of the central values and uncertainties.

## 1  Introduction

Recent measurements of $|V_{ud}|$, $|V_{us}|$ and $|V_{ub}|$ are not consistent with the unitarity condition on the first row of the CKM matrix [1,2]. Tau decay measurements are used to determine $|V_{us}|$ [3,4], supplementing the more precise determinations that are obtained using kaon decays. All these estimates also rely on lattice QCD estimates of form factors and decay constants [5,6], with the exception of the $|V_{us}|$ determinations based on the total branching fraction of the tau lepton into strange final states. Two of the $|V_{us}|$ determinations using tau measurements rely on estimates of the radiative corrections for the branching fractions $B(\tau \to \pi/K\nu)$, which are computed using

also the radiative corrections for $B(\pi \to \ell\nu)$ and $B(K \to \ell\nu)$ [7]. New estimates of these radiative corrections have been computed with a novel approach using lattice QCD+QED [8]. We evaluate in the following the impact of these new estimates on the $|V_{us}|$ determinations using tau decay measurements.

## 2   $|V_{us}|$ from $B(\tau \to K\nu)/B(\tau \to \pi\nu)$ and $B(\tau \to K\nu)$

We use the tau branching fractions of the HFLAV 2018 report fit [3] and we compute $|V_{us}|$ using the updated external inputs provided by the Review of Particle Physics [9], by the FLAG review of lattice QCD calculations [5, 6], by CODATA 2018 [10]. In updating the CODATA constants from the values used in the HFLAV 2018 report, a numerical trascription error that slightly affected this $|V_{us}|$ determination in the HFLAV 2018 report has been corrected. Finally, we obtain $|V_{us}|$ using both the original [7] and the recently published [8] radiative corrections.

We compute $|V_{us}|$ as in the HFLAV 2018 report from the ratio of branching fractions $B(\tau^- \to K^-\nu_\tau)/B(\tau^- \to \pi^-\nu_\tau)$ and from the branching fraction $B(\tau^- \to K^-\nu_\tau)$ using the equations

$$\frac{B(\tau^- \to K^-\nu_\tau)}{B(\tau^- \to \pi^-\nu_\tau)} = \frac{f_{K\pm}^2\,|V_{us}|^2}{f_{\pi\pm}^2\,|V_{ud}|^2}\frac{\left(m_\tau^2 - m_K^2\right)^2}{\left(m_\tau^2 - m_\pi^2\right)^2}\frac{1 + \delta R_{\tau/K}}{1 + \delta R_{\tau/\pi}}(1 + \delta R_{K\mu2/\pi\mu2})\,,$$

$$B(\tau^- \to K^-\nu_\tau) = \frac{G_F^2}{16\pi\hbar}f_{K\pm}^2\,|V_{us}|^2\,\tau_\tau m_\tau^3\left(1 - \frac{m_K^2}{m_\tau^2}\right)^2(1 + \delta R_{\tau/K})(1 + \delta R_{K\mu2})\,,$$

respectively. We use Refs. [11–14] to compute the radiative-correction factor

$$\frac{1 + \delta R_{\tau/K}}{1 + \delta R_{\tau/\pi}} = \frac{1 + (0.90 \pm 0.22)\%}{1 + (0.16 \pm 0.14)\%}\,.$$

$|V_{ud}| = 0.97373 \pm 0.00031$ is taken from a 2020 updated determination [1].

The radiation correction terms $\delta R_{K\mu2/\pi\mu2} = -0.69 \pm 0.17$ and $\delta R_{K\mu2} = 1.07 \pm 0.21$ are provided without the isospin-breaking corrections by Refs. [7, 15–18]. The same sources report also $\delta R_{\pi\mu2} = 1.76 \pm 0.21$. The three estimates are consistent with a correlation of 0.67 between $\delta R_{\pi\mu2}$ and $\delta R_{K\mu2}$, which is used when computing $|V_{us}|$. These radiative correction terms are used with the lattice QCD decay constants that include isospin-breaking corrections from the FLAG 2019 lattice QCD averages with $N_f = 2 + 1 + 1$: $f_{K\pm}/f_{\pi\pm} = 1.1932 \pm 0.0021$ [5, 19–22] and $f_{K\pm} = 155.7 \pm 0.3$ MeV [5, 20, 21, 23].

New estimates of the radiation correction terms inclusive of isospin-breaking corrections $\delta R'_{K\mu2/\pi\mu2} = -1.26 \pm 0.14$ and $\delta R'_{K\mu2} = 0.24 \pm 0.10$ are provided by Ref. [8]. The same source reports also $\delta R'_{\pi\mu2} = 1.53 \pm 0.19$. The three estimates are consistent with a correlation of 0.63 between $\delta R'_{\pi\mu2}$ and $\delta R'_{K\mu2}$, which originates from a correlation of 0.794 between $\delta R'_{\pi\mu2}$ the non-isospin-breaking component of $\delta R'_{K\mu2}$ [24]. These radiative correction terms are used with the isospin-symmetric lattice QCD decay constants with $N_f = 2 + 1 + 1$ provided by Ref. [8]: $f_K/f_\pi = 1.1966 \pm 0.0018$ and $f_K = 156.1 \pm 0.2$ MeV.

Table 1 reports the values of $|V_{us}|_{\tau K/\pi}$ from $B(\tau \to K\nu)/B(\tau \to \pi\nu)$ and $|V_{us}|_{\tau K}$ from $B(\tau \to K\nu)$ using both the original and the new $\pi\ell2$ and $K\ell2$ radiative corrections. The improvements in the precision of the radiative corrections results in minor improvements on the $|V_{us}|$

Table 1: $|V_{us}|$ determinations. $|V_{us}|_{\tau s}$ denotes $|V_{us}|$ obtained with the $\tau \to X_s \nu$ inclusive method using the HFLAV 2018 report [3], complemented with the updated external inputs described in the text. $|V_{us}|_{\text{uni}}$ denotes the value of $|V_{us}|$ assuming the CKM matrix unitarity, $|V_{ud}|$ from Ref. [1] and $|V_{us}|$ from Ref. [9]. The determinations of $|V_{us}|_{\tau K/\pi}$ and $|V_{us}|_{\tau K}$ using the exclusive tau branching fractions is reported using both the original and the new estimates of the $\pi\ell 2$ and $K\ell 2$ radiative corrections. $|V_{us}|_\tau$ denotes the combination of all three $|V_{us}|$ determinations with tau decay measurements. The signed difference with respect to $|V_{us}|_{\text{uni}}$ in standard deviations is reported for all $|V_{us}|$ determinations.

.

| | $|V_{us}|$ | | | |
|---|---|---|---|---|
| $|V_{us}|_{\text{uni}}$ | $0.2277 \pm 0.0013$ | $0.0\,\sigma$ | | |
| $|V_{us}|_{\tau s}$ | $0.2192 \pm 0.0019$ | $-3.6\,\sigma$ | | |
| | Cirigliano & Neufeld 2011 | | Di Carlo *et al.* 2019 | |
| $|V_{us}|_{\tau K/\pi}$ | $0.2234 \pm 0.0015$ | $-2.0\,\sigma$ | $0.2235 \pm 0.0015$ | $-2.0\,\sigma$ |
| $|V_{us}|_{\tau K}$ | $0.2226 \pm 0.0015$ | $-2.6\,\sigma$ | $0.2229 \pm 0.0014$ | $-2.6\,\sigma$ |
| $|V_{us}|_\tau$ | $0.2217 \pm 0.0013$ | $-3.2\,\sigma$ | $0.2219 \pm 0.0013$ | $-3.1\,\sigma$ |

determinations because of other larger contributing uncertainties. The changes of the central values are also minor compared with the total uncertainties. Figure 1 reports the $|V_{us}|$ determinations in this document compared with $|V_{us}|$ from the CKM matrix unitarity and $|V_{us}|$ from kaon decay measurements.

## 3   Conclusions

Recent new slightly more precise estimates of the $\pi\ell 2$ and $K\ell 2$ radiative corrections have been used to update the $|V_{us}|$ determination using the tau branching fractions of the HFLAV 2018 report, resulting in minor changes of the central values and uncertainties.

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
