# Peer review of "Updated determinations of |V$_{us}$| with tau decays using new recent estimates of radiative corrections for light-meson leptonic decay rates"

_SciPost Physics Proceedings_

## Round 2 · Referee Report · Anonymous · 2022-3-10

Report
The manuscript presents a relevant update of the $V_{us}$ determination from exclusive tau decays. I recommend its publication at SciPost after the author has considered implementing the few minor comments/suggestions described below.
I don't need to see the manuscript again.
Requested changes
1- The second formula in page 2 contains a funny character "$\hbar$". It should be a typo.
2- The $\delta R_{\tau/P}$ corrections displayed in the third formula of page 2 have been recently re-evaluated in 2107.04603, 2112.01859, where slightly larger errors are claimed. This is worth mentioning and showing how numbers change.
3- The table in page 3 displays also the average with the inclusive determination (last row), but this is not clear neither from the table caption nor from the text, where one only mentions "all three determinations" (which ones?). A reference to the inclusive method could also be appropriate.

---

## Round 3 · Author Response

Addressed reviewer comments.

---

## Round 3 · List of Changes

1. modified 2nd eq. in p.2 to indicate use of reduced Fermi constant and facilitate recognition of the units of each term

2. mentioned existence of additional recently published new radiative corrections, as requested

3. Table 1 & Figure 1 now report only Vus with tau measurements described in this document

4. Figure 1 now reports Vus with tau measurements for both discussed radiative corrections

5. captions of Table 1 and Figure 1 modified to account for content changes

---

## Editorial Decision

accepted_in_target_journal